# $i$-RevNet: Deep Invertible Networks

**Jörn-Henrik Jacobsen** [†‡]**, Arnold Smeulders** [†]**, Edouard Oyallon** [§]
[†]University of Amsterdam
`joern.jacobsen@bethgelab.org`

## ABSTRACT

It is widely believed that the success of deep convolutional networks is based on progressively discarding uninformative variability about the input with respect to the problem at hand. This is supported empirically by the difficulty of recovering images from their hidden representations, in most commonly used network architectures. In this paper we show via a one-to-one mapping that this loss of information is *not* a necessary condition to learn representations that generalize well on complicated problems, such as ImageNet. Via a cascade of homeomorphic layers, we build the $i$-RevNet, a network that can be fully inverted up to the final projection onto the classes, i.e. no information is discarded. Building an invertible architecture is difficult, for one, because the local inversion is ill-conditioned, we overcome this by providing an explicit inverse. An analysis of i-RevNets learned representations suggests an alternative explanation for the success of deep networks by a progressive contraction and linear separation with depth. To shed light on the nature of the model learned by the $i$-RevNet we reconstruct linear interpolations between natural image representations.

## 1 INTRODUCTION

A CNN may be very effective in classifying images of all sorts (He et al., 2016; Krizhevsky et al., 2012), but the cascade of linear and nonlinear operators reveals little about the contribution of the internal representation to the classification. The learning process is characterized by a steady reduction of large amounts of uninformative variability in the images while simultaneously revealing the essence of the visual class. It is widely believed that this process is based on progressively discarding uninformative variability about the input with respect to the problem at hand (Dosovitskiy & Brox, 2016; Mahendran & Vedaldi, 2016; Shwartz-Ziv & Tishby, 2017; Achille & Soatto, 2017). However, the extent to which information is discarded is lost somewhere in the intermediate nonlinear processing steps. In this paper, we aim to provide insight into the variability reduction process by proposing an invertible convolutional network, that does not discard any information about the input.

The difficulty to recover images from their hidden representations is found in many commonly used network architectures (Dosovitskiy & Brox, 2016; Mahendran & Vedaldi, 2016). This poses the question if a substantial loss of information is necessary for successful classification. We show information does not have to be discarded. By using homeomorphic layers, the invariance can be built only at the very last layer via a projection.

In Shwartz-Ziv & Tishby (2017), minimal sufficient statistics are proposed as a candidate to explain the reduction of variability. Tishby & Zaslavsky (2015) introduces the information bottleneck principle which states that an optimal representation must reduce the mutual information between an input and its representation to reduce as much uninformative variability as possible. At the same time, the network should maximize the mutual information between the desired output and its representation to effectively preserve each class from collapsing onto other classes. The effect of the information bottleneck was demonstrated on small datasets in Shwartz-Ziv & Tishby (2017); Achille & Soatto (2017).

---

[‡]Now at Bethgelab, University of Tübingen

[§]CVN, CentraleSupélec, Université Paris-Saclay ; Galen team, INRIA Saclay
SequeL team, INRIA Lille ; DI, ENS, Université PSL

However, in this work, we show it is not a necessary condition and we build a cascade of home-omorphic layers, which preserves the mutual information between input and hidden representation and shows that the loss of information can only occur at the final layer. This way we demonstrate that a loss of information can be avoided while maintaining discriminability, even for large-scale problems like ImageNet. One way to reduce variability is progressive contraction with respect to a meaningful $\ell^2$ metric in the intermediate representations.

Several works (Oyallon, 2017; Zeiler & Fergus, 2014) observed a phenomenon of progressive separation and contraction in non-invertible networks on limited datasets. Those progressive improvements can be interpreted as the creation of progressively stronger invariants for classification. Ideally, the contraction should not be too brutal to avoid removing important information from the intermediate signal. This shows that a good trade-off between discriminability and invariance has to be progressively built. In this paper, we extend some findings of Zeiler & Fergus (2014); Oyallon (2017) to ImageNet (Russakovsky et al., 2015) and, most importantly, show that a loss of information is not necessary for observing a progressive contraction.

The duality between invariance and separation of the classes is discussed in Mallat (2016). Here, intra-class variabilities are modeled as Lie groups that are processed by performing a parallel transport along those symmetries. Filters are adapted through learning to the specific bias of the dataset and avoid to contract along discriminative directions. However, using groups beyond the Euclidean case for image classification is hard. Mainly because groups associated with abstract variabilities are difficult to estimate due to their high-dimensional nature, as well as the appropriate degree of invariance required. An illustration of this framework on the Euclidean group is given by the scattering transform (Mallat, 2012), which builds invariance to small translations while being recoverable to a certain extent. In this work, we introduce a network that cannot discard any information except at the final classification stage, while we demonstrate numerically progressive contraction and separation of the signal classes.

We introduce the $i$-RevNet, an invertible deep network.[1] $i$-RevNets retain all information about the input signal in any of their intermediate representations up until the last layer. Our architecture builds upon the recently introduced RevNet (Gomez et al., 2017), where we replace the non-invertible components of the original RevNets by invertible ones. $i$-RevNets achieve the same performance on Imagenet compared to similar non-invertible RevNet and ResNet architectures (Gomez et al., 2017; He et al., 2016).

To shed light on the mechanism underlying the generalization-ability of the learned representation, we show that $i$-RevNets progressively separate and contract signals with depth. Our results are evidence for an effective reduction of variability through a contraction with a recoverable input obtained from a series of one-to-one mappings.

## 2  RELATED WORK

Several recent works show that significant information about the input images is lost with depth in successful Imagenet classification CNNs (Dosovitskiy & Brox, 2016; Mahendran & Vedaldi, 2016). To understand the loss of information, the references propose to invert the representations by means of learned or hand-engineered priors. The approximate inversions indicate increased geometric and photometric invariance with depth. Multiple other works report progressive properties of deep networks that may be linked to discarded information in the representations as well, such as linearization (Radford et al., 2015), linear separability (Zeiler & Fergus, 2014), contraction (Oyallon, 2017) and low-dimensional embeddings (Aubry & Russell, 2015). However, it is not clear from above observations if the loss of information is a necessity for the observed progressive phenomena. In this work, we show that progressive separation and contraction can be obtained while at the same time allowing an exact reconstruction of the signal.

Multiple frameworks have been introduced that permit to learn invertible representations under certain conditions. Parseval networks (Cisse et al., 2017) have been introduced to increase the robustness of learned representations with respect to adversarial attacks. In this framework, the spectrum of convolutional operators is constrained to norm 1 during learning. The linear operator is thus injective.

---

[1]Code is available at: https://github.com/jhjacobsen/pytorch-i-revnet

As a consequence, the input of Parseval networks can be recovered if but only if the built-in non-linearities are invertible as well, which is typically not the case. Bruna et al. (2013) derive conditions under which pooling representations are, but our method directly overcomes this issue. The Scattering transform (Mallat, 2012) is an example of predefined deep representation, approximately invariant to translations, that can be reconstructed when the degree of invariance specified is small. Yet, it requires a gradient descent optimization and no guarantee of convergences are known. In summary, the references make clear that invertibility requires special care in designing the architecture or special care in designing the optimization procedure. In this paper, we introduce a network, that overcomes these issues and has an exact inverse by construction.

Our main inspiration for this work is the recent reversible residual network (RevNet), introduced in Gomez et al. (2017). RevNets are in turn closely related to NICE and Real-NVP architectures (Dinh et al., 2016; 2014), which make use of constrained Jacobian determinants for generative modeling. All these architectures are similar to the lifting scheme (Sweldens, 1998) and Feistel cipher diagrams (Menezes et al., 1996), as we will show. RevNets illustrate how to build invertible ResNet-type blocks that avoid storing intermediate activations necessary for the backward pass. However, RevNets still employ multiple non-invertible operators like max-pooling and downsampling operators as part of the network. As such, RevNets are not invertible by construction. In this paper, we show how to build an invertible type of RevNet architecture that performs competitively with RevNets on Imagenet, which we call $i$-RevNet for invertible RevNet.

## 3 THE $i$-REVNET

This section introduces the general framework of the $i$-RevNet architecture and explains how to explicitly build an inverse or a left-inverse to an $i$-RevNet. Its practical implementation is discussed, and we demonstrate competitive numerical results.

### 3.1 AN INVERTIBLE ARCHITECTURE

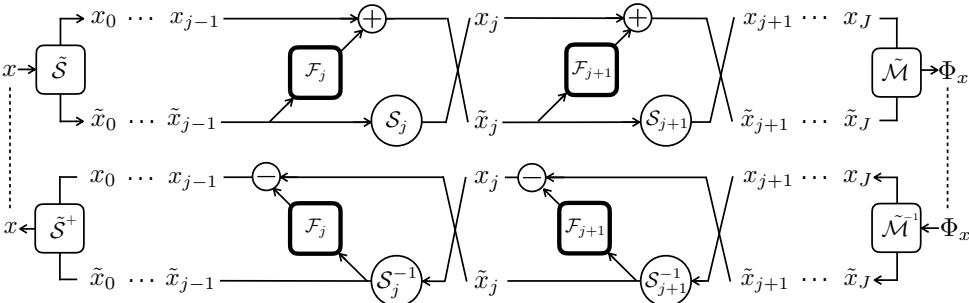

Figure 1: The main component of the $i$-RevNet and its inverse. RevNet blocks are interleaved with convolutional bottlenecks $\mathcal{F}_j$ and reshuffling operations $\mathcal{S}_j$ to ensure invertibility of the architecture and computational efficiency. The input is processed through a splitting operator $\tilde{\mathcal{S}}$, and output is merged through $\tilde{\mathcal{M}}$. Observe that the inverse network is obtained with minimal adaptations.

We describe $i$-RevNets in their general setting. Their foundations are largely grounded in the recent RevNet architecture (Gomez et al., 2017). In an $i$-RevNet, an initial input is split into two sublayers $(x_0, \tilde{x}_0)$ of equal size, thanks to a splitting operator $\tilde{\mathcal{S}}x \triangleq (x_0, \tilde{x}_0)$, in this paper we choose to split the channel dimension as is done in RevNets. The operator $\tilde{\mathcal{S}}$ is linear, injective, reduces the spatial resolution of the coefficients and can potentially increase the layer size, as wider layers usually improve the classification performance (Zagoruyko & Komodakis, 2016). We can thus build a pseudo inverse $\tilde{\mathcal{S}}^+$ that will be used for the inversion. Recall that if $\tilde{\mathcal{S}}$ is invertible, then $\tilde{\mathcal{S}}^+ = \tilde{\mathcal{S}}^{-1}$.

The number of coefficients of the next block is maintained, and at each depth $j$, the representation $\Phi_j x$ is again decoupled into two variables $\Phi_j x \triangleq (x_j, \tilde{x}_j)$ that play interlaced roles.

The strategy implemented by an $i$-RevNet consists in an alternation between additions, and non-linear operators $\mathcal{F}_j$, while progressively down-sampling the signal thanks to the operators $\mathcal{S}_j$. Here, $\mathcal{F}_j$ consists of convolutions and non-linearity on $\tilde{x}_j$. The pair of the final layer is concatenated through a merging operator $\tilde{\mathcal{M}}$. We will omit $\tilde{\mathcal{M}}, \tilde{\mathcal{M}}^{-1}, \tilde{\mathcal{S}}^+$ and $\tilde{\mathcal{S}}$ for the sake of simplicity, when not necessary. Figure 1 describes the blocks of an $i$-RevNet. The design is similar to the Feistel cipher diagrams (Menezes et al., 1996) or a lifting scheme (Sweldens, 1998), which are invertible and efficient implementations of complex transforms like second generation wavelets.

In this way, we avoid the non-invertible modules of a RevNet (e.g. max-pooling or strides) which are necessary to train them in a reasonable time and are designed to build invariance w.r.t. translation variability. Our method shows we can replace them by linear and invertible modules $\mathcal{S}_j$, that can reduce the spatial resolution (we refer to it as a spatial down-sampling for the sake of simplicity) while maintaining the layer's size by increasing the number of channels.

We keep the computational cost manageable by tightly coupling downsampling and increase in width of the network. Reducing the spatial resolution can be undesirable, so $\mathcal{S}_j$ can potentially be the identity. We refer to such networks as $i$-RevNets. This leads to the following equations:

$$\begin{cases} x_{j+1} = \mathcal{S}_{j+1}\tilde{x}_j \\ \tilde{x}_{j+1} = x_j + \mathcal{F}_{j+1}\tilde{x}_j \end{cases} \iff \begin{cases} \tilde{x}_j = \mathcal{S}_{j+1}^{-1} x_{j+1} \\ x_j = \tilde{x}_{j+1} - \mathcal{F}_{j+1}\tilde{x}_j \end{cases} \tag{1}$$

Our downsampling layer can be written for $u$ the spatial variable and $\lambda$ the channel index:

$$\mathcal{S}_j x(u, \lambda) = x(\Psi(u, \lambda))$$

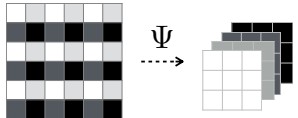

where $\Psi$ is some invertible mapping. In principle, any invertible downsampling operation like e.g. dilated convolutions (Yu & Koltun, 2015) can be considered here. We use the inverse of the operation described in Shi et al. (2016) as illustrated in Figure 2, since it preserves roughly the spatial ordering, and thus permits to avoid mixing different neighborhoods via the next convolution. $\tilde{\mathcal{S}}$ is similar, but also linearly increases the channel dimensionality, for example by concatenating 0.

Figure 2: Illustration of the invertible down-sampling

The final layer $\Phi x \triangleq \Phi_J x = (x_J, \tilde{x}_J)$ is then averaged along the spatial dimension, followed by a ReLU non-linearity and finally a linear projection on the class probes, which are fed to a supervised training algorithm. From a given $i$-RevNet, it is possible to define a left-inverse $\Phi^+$, i.e. $\Phi^+\Phi x = x$ or even an inverse $\Phi^{-1}$, i.e. $\Phi^{-1}\Phi x = \Phi^{-1}\Phi x = x$ if $\tilde{\mathcal{S}}$ is invertible. In these cases, the convolutional sections are as well some $i$-RevNets. An $i$-RevNet is the dual of its inverse, in the sense that it requires to replace $(\mathcal{S}_j, \mathcal{F}_j)$ by $(\mathcal{S}_j^{-1}, -\mathcal{F}_j)$ at each depth $j$, and to apply $\tilde{\mathcal{S}}^+$ on the output. In consequence, its implementation is simple and specified by Equation (1). In Subsection 4.2, we discuss that the inverse of $\Phi$ does not suffer from significant round-off errors, while however being very sensitive to small variations of an input on a large subspace, as shown in Subsection 4.1.

## 3.2 ARCHITECTURE, TRAINING AND PERFORMANCES

In this subsection, we describe two models that we trained: an injective $i$-RevNet (a) and a bijective $i$-RevNet (b), with fewer parameters. The hyper-parameters were selected to be either close to the ResNet and RevNet baselines in terms of the number of layers (a) or parameters (b) while keeping performance competitive. For the same reasons as in Gomez et al. (2017), our scheme also allows avoiding storing any intermediate activations at training time, making memory consumption for very deep $i$-RevNets not an issue in practice. We compare our implementation with a RevNet with 56 layers corresponding to $28M$ parameters, as provided in the open source release of Gomez et al. (2017), and with a standard ResNet of 50 layers, with $26M$ parameters (He et al., 2016).

Each block $\mathcal{F}_j$ is a bottleneck block, which consists of a succession of 3 convolutional operators, each preceded by Batchnormalization (Ioffe & Szegedy, 2015) and ReLU non-linearity. The second layer has four times fewer channels than the other two, while their corresponding kernel sizes are respectively $1 \times 1, 3 \times 3, 1 \times 1$.

| Architecture | Injective | Bijective | Top-1 error | Parameters |
|---|---|---|---|---|
| ResNet | - | - | 24.7 | 26M |
| RevNet | - | - | 25.2 | 28M |
| $i$-RevNet (a) | yes | - | 24.7 | 181M |
| $i$-RevNet (b) | yes | yes | 26.7 | 29M |

Table 1: Comparison of different architectures trained on ILSVRC-2012, in terms of classification accuracy and number of parameters

The final representation is spatially averaged and projected onto the 1000 classes after a ReLU non-linearity. We now discuss how we progressively decrease the spatial resolution, while increasing the number of channels per layer by use of the operators $\mathcal{S}_j$.

We first describe the model (a), that consists of 56 layers which have been optimized to match the performances of a RevNet or a ResNet with approximatively the same number of layers. In particular, we explain how we progressively decrease the spatial resolution, while increasing the number of channels per block by use of the operators $\mathcal{S}_j$.

The splitting operator $\tilde{\mathcal{S}}$ consists in a linear and injective embedding that downsamples by a factor $4^2$ the spatial resolution by increasing the number of output channels from 48 to 96 by simply adding 0. The latter permits to increase the initial layer size, and consequently, the size of the next layers as performed in Gomez et al. (2017); it is thus not a bijective yet an injective $i$-RevNet. At depth $j$, $\mathcal{S}_j$ allows us to reduce the number of computations while maintaining good classification performance. It will correspond to a downsampling operator respectively at the depth $3j = 15, 27, 45$ (3j as one block corresponds to three layers), similar to a normal RevNet. The spatial resolution of these layers is reduced by a factor $2^2$ while increasing the number of channels by a factor of 4 respectively to 48, 192, 768 and 3072. Furthermore, it means that the corresponding spatial resolutions for an input of size $224^2$ are respectively $112^2, 56^2, 28^2, 14^2, 7^2$. The total number of coefficients at each layer is then about $0.3M$. All the remaining blocks $\mathcal{S}_j$ are kept fix to the identity as explained in the section above.

Architecture (b) is bijective, it consists of 300 layers (100 blocks), whose total numbers of parameters have been optimized to match those of a RevNet with 56 layers. Initially, the input is split via $\tilde{\mathcal{S}}$, which corresponds to an invertible spatial downsampling of $2^2$ that increases the number of channels from 3 to 12. It thus keeps the dimension constant and permits building a bijective $i$-RevNet. Then, at depth $3j = 3, 21, 69, 285$, the spatial resolution is reduced by $2^2$ via $\mathcal{S}_j$. Contrary to the architecture (a), the dimensionality of each layer is constantly equal to $3 \times 224^2$, until the final layer, with channel sizes of 24, 96, 384, 1536.

For both networks, the training on Imagenet follows the same setup as Gomez et al. (2017). We train with SGD and momentum of 0.9. We regularized the model with a $\ell^2$ weight decay of $10^{-4}$ and batch normalization. The dataset is processed for 600k iterations on a batch size of 256, distributed on 4GPUs. The initial learning rate is 0.1, dropped by a factor of ten every 160k iterations. The dataset was augmented according to Gomez et al. (2017). The images values are mapped to $[0, 1]$ while following geometric transformations were applied: random scaling, random horizontal flipping, random cropping of size $224^2$, and finally color distortions. No other regularizations were incorporated into the classification pipeline. At test time, we rescale the image size to $256^2$ and perform a center crop of size $224^2$.

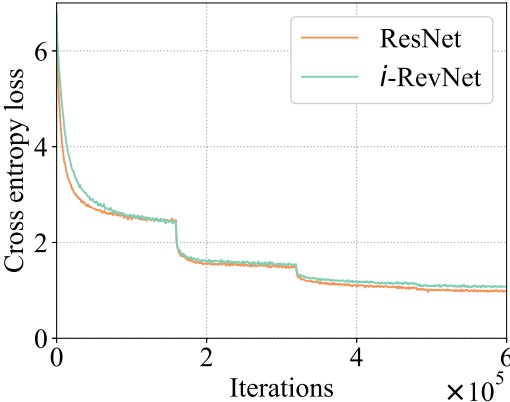

Figure 3: Training loss of the $i$-RevNet (b), compared to the ResNet, on ImageNet.

We report the training loss (i.e. Cross entropy) curves in Figure 3 of our $i$-RevNet (b) and the ResNet baseline, displayed is a moving average over 100 iterations. Observe that the decrease of both training-losses are very similar which indicates that the constraint of invertibility does not interfere negatively with the learning process. However, we observed one third longer wall-clock times for $i$-RevNets compared to plain RevNets because the channel size becomes larger. The Table 1 reports the performances of our $i$-RevNets, with comparable RevNet and ResNet. First, we compare the $i$-RevNet (a) with the RevNet and ResNet. Indeed, those CNNs have the same number of layers, and the $i$-RevNet (a) increases the channel width of the initial layer as done in Gomez et al. (2017). The drawback of this technique is that the kernel sizes will be larger for all subsequent layers.

The $i$-RevNet (a) has about 6 times more parameters than a RevNet and a ResNet but leads to a similar accuracy on the validation set of ImageNet. On the contrary, the $i$-RevNet (b) is designed to have roughly the same number of parameters as the RevNet and ResNet, while being bijective. Its accuracy decreases by 1.5% absolute percent on ImageNet compared to the RevNet baseline, which is not surprising because the number of channels was not drastically increased in the earlier layers as done in the baselines (Gomez et al., 2017; Krizhevsky et al., 2012; He et al., 2016); we did not explore wide ranges of hyper-parameters, thus the gap between (a) and (b) can likely be reduced with additional engineering.

## 4 ANALYSIS OF THE INVERSE

We now analyze the representation $\Phi$ built by our bijective neural network $i$-RevNet (b) and its inverse $\Phi^{-1}$, as trained on ILSVRC-2012. We first explain why obtaining $\Phi^{-1}$ is challenging, even locally. We then discuss the reconstruction, while displaying in the image space linear interpolations between representations.

### 4.1 AN ILL-CONDITIONED INVERSION

In the previous section, we have described the $i$-RevNet architecture, that permits defining a deep network with an explicit inverse. We explain now why this is normally difficult, by studying its local inversion. We study the local stability of a network $\Phi$ and its inverse $\Phi^{-1}$ w.r.t. to its input, which means that we will quantify locally the variations of the network and its inverse w.r.t. to small variations of an input. As $\Phi$ is differentiable (and its inverse as well), an equivalent way to perform this study is to analyze the singular values of the differential $\partial\Phi$ at some point, as for $(a, b)$ close the following holds:

$$\Phi a \approx \Phi b + \partial\Phi_b(a - b).$$

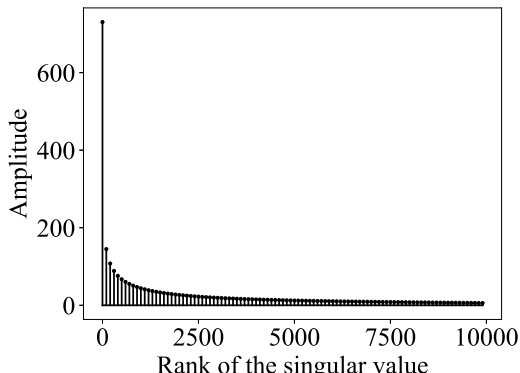

Figure 4: Normalized sorted singular values of $\partial\Phi_x$.

Ideally, a well-conditioned operator has all its singular values constant equal to 1, for instance as achieved by the isometric operators of Cisse et al. (2017).

In our numerical application to an image $x$, $\partial\Phi_x$ corresponds to a very large matrix (square of the number of coefficients of the image at least) whose computations are expensive. Figure 4 corresponds to the singular values of the differential (i.e. the square roots of the eigen values of $\partial\Phi^*\partial\Phi$), in decreasing order, for a given natural image from ImageNet. The example we plot is typical of the behavior of $\partial\Phi$. Observe there is a fast decay: numerically, the first $10^3$ and $10^4$ singular values are responsible respectively for $80\%$ and $97\%$ of the cumulated energy (i.e. sum of squared singular values). This indicates $\Phi$ linearizes the space locally in a considerably smaller space in comparison to the original input dimension. However, the dimensionality is still quite large (i.e. $> 10$) and thus we can not infer that $\Phi$ lays locally in a low-dimensional manifold. It also proves that inversing $\Phi$ is difficult and is an ill-conditioned problem. Thus obtaining implicitly this inverse would be a challenging task that we avoided, thanks to the formal reconstruction algorithm provided by Subsection 3.1.

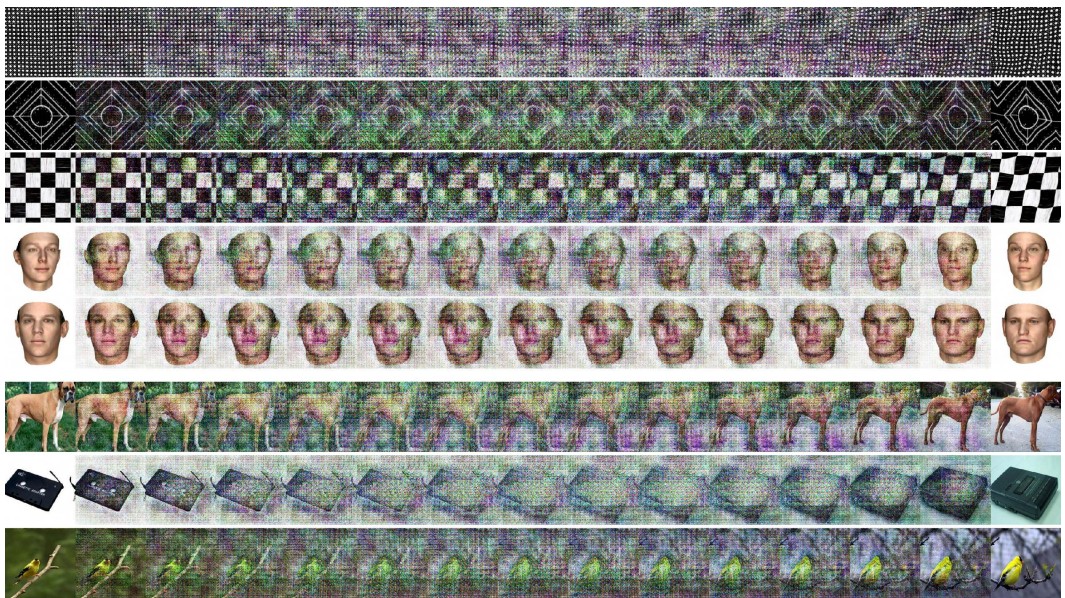

Figure 5: This graphic displays several reconstructed sequences $\{x^t\}_t$. The left image corresponds to $x^0$ and the right image to $x^1$.

## 4.2 LINEAR INTERPOLATION AND RECONSTRUCTION

Visualizing or understanding the important directions in the representation of inner layers of a CNN, and in particular, the final layer is complex because typically the cascade is either not invertible or unstable. One approach to reconstruct from an output layer consists in finding the input image that matches the activation through via gradient descent. However, this technique leads only to a partial or informal reconstruction (Mahendran & Vedaldi, 2015).

Another method consists in embedding the representation in a lower dimensional space and comparing the common attributes of nearest neighbors (Szegedy et al., 2013). It is also possible to train a CNN to reconstruct the representation (Dosovitskiy & Brox, 2016). Yet these methods require a priori knowledge in order to find the appropriate embeddings or training sets. We now discuss the improvements achieved by the $i$-RevNet.

Our main claim is that while the local inversion is ill-conditioned, the inverse $\Phi^{-1}$ computations do not involve significant round-off errors. The forward pass of the network does not seem to suffer from significant instabilities, thus it seems coherent to assume that this will hold for $\Phi^{-1}$ as well. For example, adding constraints beyond vanishing moments in the case of a Lifting scheme is difficult (Sweldens, 1998; Mallat, 1999), and this is a weakness of this method. We validate our claim by computing the empirical relative error on several subsets $\mathcal{X}$ of data:

$$\epsilon(\mathcal{X}) = \frac{1}{|\mathcal{X}|} \sum_{x \in \mathcal{X}} \frac{\|x - \Phi^{-1}\Phi x\|}{\|x\|}$$

We evaluate this measure on a subset $\mathcal{X}_1$ of $|\mathcal{X}_1| = 10^4$ independent uniform noises and on the validation set $\mathcal{X}_2$ of ImageNet. We report $\epsilon(\mathcal{X}_1) = 5 \times 10^{-6}$ and $\epsilon(\mathcal{X}_2) = 3 \times 10^{-6}$ respectively, which are close to the machine error and indicates that the inversion does not suffer from significant round-off errors.

Given a pair of images $\{x^0, x^1\}$, we propose to study linear interpolations between the pair of representations $\{\Phi x^0, \Phi x^1\}$, in the feature domain. Those interpolations correspond to existing images as $\Phi^{-1}$ is an exact inverse. We reconstruct a convex path between two input points; it means that if:

$$\phi^t = t\Phi x^0 + (1 - t)\Phi x^1,$$

then: $x^t = \Phi^{-1}\phi^t$ is a signal that corresponds to an image.

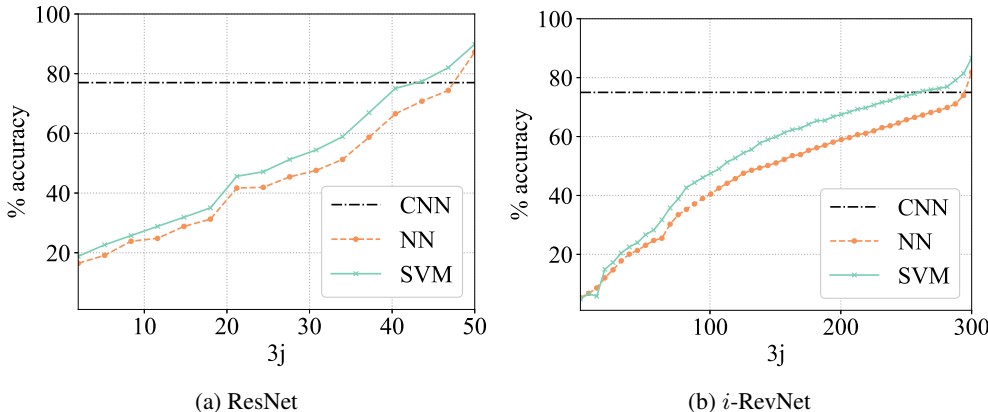

(a) ResNet                    (b) $i$-RevNet

Figure 6: Accuracy at depth $j$ for a linear SVM and a 1-nearest neighbor classifier applied to the spatially averaged $\Phi_j$.

We discretized $[0, 1]$ into $\{t_1, ..., t_k\}$, adapt the step size manually and reconstruct the sequence of $\{x^{t_1}, ..., x^{t_k}\}$. Results are displayed in the Figure 5. We selected images from the basel face dataset (Paysan et al., 2009), describable texture dataset (Cimpoi et al., 2014) and imagenet.

We now interpret the results. First, observe that a linear interpolation in the feature space is not a linear interpolation in the image space and that intermediary images are noisy, even for small deformations, yet they mostly remain recognizable. However, some geometric transformations such as a 3D-rotation seem to have been linearized, as suggested in Aubry & Russell (2015). In the next section, we thus investigate how the linear separation progresses with depth.

## 5  A CONTRACTION

In this section, we study again the bijective $i$-RevNet. We first show that a localized or linear classifier progressively improves with depth. Then, we describe the linear subspace spanned by $\Phi$, namely the *feature space*, showing that the classification can be performed on a much smaller subspace, which can be built via a PCA.

### 5.1  PROGRESSIVE LINEAR SEPARATION AND CONTRACTION

We show that both a ResNet and an $i$-RevNet build a progressively more linearly separable and contracted representation as measured in Oyallon (2017). Observe this property holds for the $i$-RevNet despite the fact that it can not discard any information.

We investigate these properties in each block, with the following experimental protocol. To reduce the computational burden we used a subset of 100 randomly selected imagenet classes, that consist of $N = 120k$ images, and keep the same subset during all our following experiments. At each depth $j$, we extract the features $\{\Phi_j x^n\}_{n \leq N}$ of the training set, we average them along the spatial variable and standardize them in order to avoid any ill-conditioning effects. We used both a nearest neighbor classifier and a linear SVM. The former is a localized classifier that indicates that the $\ell^2$ metric is progressively more important for classification, while a linear SVM measures the linear separation of the different classes. The parameters of the linear SVM are cross-validated on a small subset of the training set, prior to training on the 100 classes. We evaluate both classifiers for each model on the validation set of ImageNet and report the Top-1 accuracy in Figure 6.

We observe that both classifiers progressively improve similarly with depth for each model, the linear SVM performing slightly better than the nearest neighbor classifier because it is the more robust and discriminative classifier of the two. In the case of the $i$-RevNet, the classification performed by the CNN leads to 77%, and the linear SVM performs slightly better because we did not fine-tune the model to 100 classes. Observe that there is a more intense jump of performance on the 3 last layers, which seems to indicate that the former layers have prepared the representation to be more contracted and linearly separated for the final layers.

The results suggest a low-dimensional embedding of the data, but this is difficult to validate as estimating local dimensionality in high dimensions is an open problem. However, in the next section, we try to compute the dimension of the discriminative part of the representation built by an $i$-RevNet.

## 5.2 DIMENSIONALITY ANALYSIS OF THE FEATURE SPACE

In this section, we investigate if we can refine the dimensionality of informative variabilities in the final layer of an $i$-RevNet. Indeed, the cascade of convolutional operators has been trained on the training set to separate the 1000 different classes while being a homeomorphism on its feature space. Thus, the dimensionality of the feature space is potentially large.

As shown in the previous subsection, the final layer is progressively prepared to be projected on the final probes corresponding to the classes. This indicates that the non-informative variabilities for classification can be removed via a linear projection on the final layer $\Phi$, which lie in a space of dimension 1000, at most. However, this projection has been built via supervision, which can still retain directions that have been contracted and thus will not be selected by an algorithm such as PCA. We show in fact a PCA retains the necessary information for classification in a small subspace.

To do so, we build the linear projectors $\pi_d$ on the subspace of the $d$ first principal components, and we propose to measure the classification power of the projected representation with a supervised classifier, e.g. nearest neighbor or a linear SVM, on the previous 100 class task. Again, the feature representation $\{\Phi x^n\}_{n \leq N}$ are spatially averaged to remove the translation variability, and standardized on the training set. We apply both classifiers, and we report the classification accuracy of $\{\pi_d \Phi x^n\}_{n \leq N}$ w.r.t. to $d$ on the Figure 7. A linear projection removes some information that can not be recovered by a linear classifier, therefore we observe that the classification accuracy only decreases significantly for $d \leq 200$. This shows that the signal indeed lies in a subspace much lower dimensional than the original feature dimensions

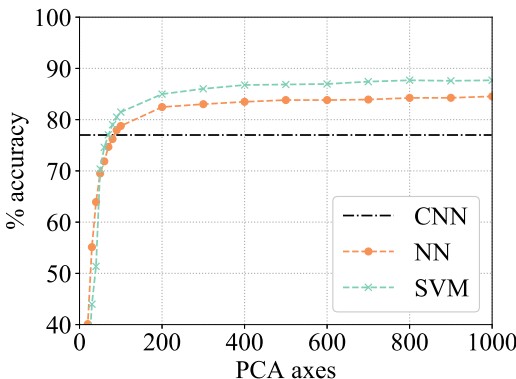

Figure 7: Accuracy of a linear SVM and nearest neighbor against the number of principal components retained.

that can be extracted simply with a PCA that only considers directions of largest variances, illustrating a successful contraction of the representation.

## 6 CONCLUSION

Invertible representations and their relationship to loss of information are on the agenda of deep learning for some time. Understanding how transformations in feature space are related to the corresponding input is an important step towards interpretable deep networks, invertible deep networks may play an important role in such analysis since, for example, one could potentially back-track a property from the feature space to the input space. To the best of our knowledge, this work provides the first empirical evidence that learning invertible representations that do not discard any information about their input on large-scale supervised problems is possible.

To achieve this we introduce the $i$-RevNet class of CNN which is fully invertible and permits to exactly recover the input from its last convolutional layer. $i$-RevNets achieve the same classification accuracy in the classification of complex datasets as illustrated on ILSVRC-2012, when compared to the RevNet (Gomez et al., 2017) and ResNet (He et al., 2016) architectures with a similar number of layers. Furthermore, the inverse network is obtained for free when training an $i$-RevNet, requiring only minimal adaption to recover inputs from the hidden representations.

The absence of loss of information is surprising, given the wide believe, that discarding information is essential for learning representations that generalize well to unseen data. We show that this is not the case and propose to explain the generalization property with empirical evidence of progressive separation and contraction with depth, on ImageNet.

Acknowledgements

Jörn-Henrik Jacobsen was partially funded by the STW perspective program ImaGene. Edouard Oyallon was partially funded by the ERC grant InvariantClass 320959, via a grant for PhD Students of the Conseil régional dIle-de-France (RDM-IdF), and a postdoctoral grant from the from DPEI of Inria (AAR 2017POD057) for the collaboration with CWI. We thank Berkay Kicanaoglu for the Basel Face data, Mathieu Andreux, Eugene Belilovsky, Amal Rannen, Patrick Putzky and Kyriacos Shiarlis for feedback on drafts of the paper.

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
