# OpenReview forum: "i-RevNet: Deep Invertible Networks"
_ICLR.cc/2018/Conference — Accept (Poster)_

### Official Review · AnonReviewer1 · 2017-11-27
**The paper proposes an invertible architecture which allows to verify that the loss of input information is not needed in deep architectures in order to generalize on large scale supervised tasks.**

**Rating:** 9
**Confidence:** 4

**Review:**

In this paper, the authors propose deep architecture that preserves mutual information between the input and the hidden representation and show that the loss of information can only occur at the final layer. They illustrate empirically that the loss of information can be avoided on large-scale classification such as ImageNet and propose to build an invertible deep network that is capable of retaining the information of the input signal through all the layers of the network until the last layer where the input could be reconstructed.

The authors demonstrate that progressive contraction and separation of the information can be obtained while at the same time allowing an exact reconstruction of the signal.

As it requires a special care to design an invertible architecture, the authors architecture is based on the recent reversible residual network (RevNet) introduced in (Gomez et al., 2017) and an invertible down-sampling operator introduced in (Shi et al., 2016). The inverse (classification) path of the network uses the same convolutions as the forward (reconstructing) one. It also uses subtraction operations instead of additions in the output computation in order to reconstruct intermediate and input layers.

To show the effectiveness of their approach on large-scale classification problem, the authors report top-1 error rates on the validation set of ILSVRC-2012. The obtained result is competitive with the original Resnet and the RevNet models. However, the proposed approach is expensive in terms of parameter budget as it requires almost 6.5 times more parameters than the RevNet and the Resnet architectures. Still, the classification and the reconstructing results are quite impressive as the work is the first empirical evidence that learning invertible representation that preserves information about the input is possible on large-scale classification tasks. Worth noting that recently, (Shwartz-Ziv and Tishby) demonstrated, not on large-scale datasets but on small ones, that an optimal representation for a classification task must reduce as much uninformative variability as possible while maximizing the mutual information between the desired output and its representation in order discriminate as much as possible between classes. This is called “information bottleneck principle”. The submitted paper shows that this principle is not a necessary condition large-scale classification.

The proposed approach is potentially of great benefit. It is also simple and easy to understand. The paper is well written and the authors position their work with respect to what has been done before. The spectral analysis of the differential operator in section 4.1 provide another motivation for the “hard-constrained” invertible architecture. Section 4.2 illustrates the ability of the network to reconstruct input signals. The visualization obtained suggests that network performs linear separation between complex learned factors. Section 5 shows that even when using either an SVM or a Nearest Neighbor classifier on n extracted features from a layer in the network, both classifiers progressively improve with deeper layers. When the d first principal components are used to summarize the n extracted features, the SVM and NN classifier performs better when d is bigger. This shows that the deeper the network gets, the more linearly separable and contracted the learned representations are.

In the conclusion, the authors state the following: “The absence of loss of information is surprising, given the wide believe, that discarding information is essential for learning representations that generalize well to unseen data”. Indeed, the authors have succeed in showing that this is not necessarily the case. However, the loss of information might be necessary to generalize well on unseen data and at the same time minimize the parameter budget for a given classification task.

---

> ### Author Response · Authors · 2017-12-28
> **Reply AnonReviewer1**
>
> We thank the reviewer very much for this encouraging review and the comments on our paper.
> We would also like to thank the reviewer for acknowledging the presented results being impressive and potentially of great benefit.
>
> Inspired by the reviewer's remark on model size and the quest for an optimal parameter budget, we have added another model to the paper that has a similar number of parameters as the RevNet and ResNet baselines. This way we show that an increased number of parameters is not necessary to obtain the invertible architecture. This newly added i-RevNet replaces the initial injective mapping with a bijective mapping. In consequence, the new model is slightly different in architecture from the baselines, as it keeps the input dimensionality constant. We have replaced the analysis of the injective i-RevNet by an analysis of the bijective i-RevNet throughout the whole paper.
>
> Furthermore, to show that the observed separation and contraction occur independently of invertibility, we have added a non-invertible ResNet baseline to the model analysis in section 5.1. We have also added training plots of ResNets compared to i-RevNets. The results show a progressive separation and contraction in invertible and non-invertible models and very similar training behaviour.
>
> Our main conclusions remain the same, while the new results substantiate their generality.
>
> We thank the reviewer once again for the insightful comments thanks to which we were able to further improve the paper.

---

### Official Review · AnonReviewer2 · 2017-11-27
**Gives an interesting insight that loss of information is not necessary for good generalizable features.**

**Rating:** 8
**Confidence:** 4

**Review:**

ICLR I-Revnet


This paper build on top of ReVNets (Gomez et al., 2017)  and introduce a variant that is fully
invertible. The model performs comparable to its variants without any loss of information.
They analyze the model and its learned representations from multiple perspectives in detail.

It is indeed very interesting an thought provoking to see that contrary to popular belief in the community no information loss is necessary to learn good generalizable features. What is missing, is more motivation for why such a property is desirable. As the authors mentions the model size has almost doubled compared to comparable ResNet. And the study of the property of the learned futures might probably limited to this i-RevNet only. It would be good to see more motivation, beside the valuable insight of knowing it’s possible.

Generally the paper is well written and readable, but few minor comments:
1-Better formatting such as putting results in model sizes, etc in tables will make them easier to find.
2-Writing down more in detail 3.1, ideally in algorithm or equation than all in text as makes it hard to read in current format.

---

> ### Author Response · Authors · 2017-12-28
> **Reply AnonReviewer2**
>
> We thank the reviewer very much for the valuable comments and for acknowledging that our main claims are very interesting and thought-provoking. In the following, we will elaborate on the increased model size and usefulness of such an architecture in detail.
>
> To add another dimension to the model analysis and to shed light on the necessary model size, we have added a model which replaces the initial injective operator with a bijective operator as used in later layers. This model has almost the same number of parameters as the baselines and trains about a day faster, albeit performs worse by 1.5%. This is to show, that model size can be reduced substantially while the invertibility property improves.
>
> ==> The authors mention model size has almost doubled
>
> Thanks to this important remark, we have added another model that shows it is possible to avoid an excessive increase of model size in i-RevNets. The newly added model has 29M parameters as opposed to 28M in the RevNet baseline while having a top-1 accuracy of 73.3%, which is ~1.5% worse than the RevNet baseline.
>
> We thank the reviewer once again for raising this point and believe that the newly introduced model makes the paper even stronger, as it shows that the invertibility property can even be improved by decreasing model size.
>
> ==> Does the analysis apply to other models as well?
>
> We thank the reviewer for this question, section 5.1 shows that progressive properties that are known to hold for lossy AlexNet type models on limited datasets, are in fact also possible to obtain in an architecture that is not able to discard information about the input on a large-scale task like Imagenet.
>
> To further strengthen the results, we have extended our analysis of the separation contraction to a ResNet baseline. Our results show, that the behaviour of the non-invertible ResNet is the same as the one observed in i-RevNets, substantiating the generality of our findings.
>
> ==> Why is such a model desirable?
>
> The core question we answer is if the success of deep convolutional networks is based on progressively discarding uninformative variability, which is a wide standing believe in the CV and ML community. We show this does not have to be the case, which has been acknowledged as "important", "interesting" and "thought-provoking" by all reviewers. Thus, the invertibility property is desirable for understanding the success of deep learning better and shed light on some of the necessities for it to work well.
> From a practical point of view, invertible models are useful for feature visualization [1,2,3] and possibly useful to overcome difficulties in upsampling/decoding pixel-wise tasks that are still quite challenging [4]. Further, lossless models might be a good candidate for transfer learning.
>
> In summary, we do believe that besides the theoretical interest of our work, which has been acknowledged by all reviewers, there is also a potential impact in deep learning applications for invertible models.
>
> We thank the reviewer once again for the important questions and remarks, we believe that the added discussion and results of the new bijective i-RevNet and ResNet baseline substantially improve the paper. We have also incorporated suggested formatting improvements into the manuscript.
>
> [1] Mahendran, Aravindh, and Andrea Vedaldi. "Understanding deep image representations by inverting them." Proceedings of the IEEE conference on computer vision and pattern recognition. 2015.
> [2] Dosovitskiy, Alexey, and Thomas Brox. "Inverting visual representations with convolutional networks." Proceedings of the IEEE Conference on Computer Vision and Pattern Recognition. 2016.
> APA
> [3] Selvaraju, Ramprasaath R., et al. "Grad-cam: Why did you say that? visual explanations from deep networks via gradient-based localization." arXiv preprint arXiv:1610.02391 (2016).
> [4] Wojna, Zbigniew, et al. "The Devil is in the Decoder." arXiv preprint arXiv:1707.05847 (2017).

---

> > ### Comment · AnonReviewer2 · 2018-01-12
> > **Response.**
> >
> > Thank you for the detail response and adding new sets of experiments for the questions that was raised and I updated the score to reflect the changes.

---

### Official Review · AnonReviewer3 · 2017-11-27

**Rating:** 8
**Confidence:** 4

**Review:**



The paper is well written and easy to follow. The main contribution is to propose a variant of the RevNet architecture that has a built in pseudo-inverse, allowing for easy inversion. The results are very surprising in my view: the proposed architecture is nearly invertible and is able to achieve similar performance as highly competitive variants: ResNets and RevNets.

The main contribution is to use linear and invertible operators (pixel shuffle) for performing downsampling, instead of non-invertible variants like spatial pooling. While the change is small, conceptually is very important.

Could you please comment on the training time? Although this is not the point of the paper, it would be very informative to include learning curves. Maybe discarding information is not essential for learning (which is surprising), but the cost of not doing so is payed in learning time. Stating this trade-off would be informative. If I understand correctly, the training runs for about 150 epochs, which is maybe double of what the baseline ResNet would require?

The authors evaluate in Section 4.2 the show samples obtained by the pseudo inverse and study the properties of the representations learned by the model. I find this section really interesting. Further analysis will make the paper stronger.

Are the images used for the interpolation train or test images?

I assume that the network evaluated with the Basel Faces dataset, is the same one trained on Imagenet, is that the case?

In particular, it would be interesting (not required) to evaluate if the learned representation is able to linearize a variety of geometric image transformations in a controlled setting as done in:

Hénaff, O,, and Simoncelli, E. "Geodesics of learned representations." arXiv preprint arXiv:1511.06394 (2015).

Could you please clarify, what do you mean with fine tuning the last layer with dropout?

The authors should cite the work on learning invertible functions with tractable Jacobian determinant (and exact and tractable log-likelihood evaluation) for generative modeling. Clearly the goals are different, but nevertheless very related. Specifically,

Dinh, L. et al  "NICE: Non-linear independent components estimation." arXiv preprint arXiv:1410.8516 (2014).


Dinh, L. et al "Density estimation using Real NVP." arXiv preprint arXiv:1605.08803 (2016).

The authors mention that the forward pass of the network does not seem to suffer from significant instabilities. It would be very good to empirically evaluate this claim.

---

> ### Author Response · Authors · 2017-12-28
> **Reply AnonReviewer3**
>
> We thank the reviewer very much for raising many interesting and important points. Furthermore, we thank the reviewer for acknowledging that the presented results are surprising and our technical contributions conceptually important. We are also pleased that the reviewer finds the analysis of the learned representation very interesting.
>
> To open up another dimension of the analysis, we have added a model which replaces the initial injective operator with a bijective operator as used in later layers. This model has almost the same number of parameters as the baseline and trains about a day faster, albeit performs worse by 1.5%. This is to show, that model size can be reduced substantially while the invertibility property improves.
>
>  ==> Maybe discarding information is not essential for learning (which is surprising), but the cost of not doing so is paid in learning time.
>
> Thank you for raising this interesting point.
> We have added plots of the loss curves to the paper that show very similar training behaviour for an i-RevNet compared to a non-invertible ResNet baseline. Training hyperparameters (e.g. learning rate schedule, training iterations, regularization) are identical for all models we have analyzed in the paper.
> Thus, there does not seem to be a cost to pay for not discarding information in terms of convergence behaviour.
>
> ==> Are the images used for the interpolation train or test images?
>
> The images used for interpolation are partially from datasets not seen during training (describable textures, Basel faces) and from the Imagenet training set. All interpolations have been obtained from the ILSVRC-2012 trained model.
>
> ==> Could you please clarify, what do you mean with fine tuning the last layer with dropout?
>
> We thank the reviewer for raising this question. For the sake of brevity, we have removed this fine-tuning in the current revision entirely. This change only affected Figure 4, we have updated the figure with interpolations from the newly added bijective model.
>
> ==> The authors mention that the forward pass of the network does not seem to suffer from significant instabilities. It would be very good to empirically evaluate this claim.
>
> Thank you for this important remark, we have empirically evaluated our claims by measuring the normalized l2 error between original and reconstruction on the whole validation set of ILSVRC-2012 and on randomly drawn uniform noise \in [-1,1], with the same number of draws as the size of the validation set. We report expectation of the error over all samples. The results show no significant instabilities and the error is visually imperceivable:
>
> i-RevNet bijective:
> ILSVRC-2012 validation set reconstruction error: 5.17e-06
> 50k uniform noise draws reconstruction error: 2.63e-06
>
> i-RevNet injective:
> ILSVRC-2012 validation set reconstruction error: 8.26e-7
> 50k uniform noise draws reconstruction error: 5.52e-07
>
> We thank the reviewer for the additional references, we have added NICE and Real-NVP to the related work section and discussed their relationship to our work.
>
> To add results on more controlled geometric transformations, we have added interpolations between small geometrical perturbations to the reconstruction experiment.
>
> We thank the reviewer once again for the very interesting and important remarks. We believe they have substantially improved the manuscript and helped to clarify many important points.

---

### Author Response · Authors · 2017-12-28
**Revision Uploaded**

Dear Reviewers,

We sincerely thank you for your work and effort in reviewing the manuscript. Your comments and remarks have been very helpful to improve the quality of the paper.

We have made two major additions to the manuscript that substantiate the generality of our claims.

1) A bijective i-RevNet with 6 times fewer parameters, that shows a reduction of parameters can even improve the invertibility property.
2) An Imagenet-trained ResNet to show our findings apply to non-invertible state-of-the-art models as well.

Please find detailed answers below, and thank you once again!

---

### Public Comment · (anonymous) · 2018-01-07
**Question Regarding Details of Downsampling for Bijective i-RevNet**

I have a question regarding the description of S_j in section 3.2. It's mentioned that "at each layer of depth j = 3, 21, 69, 285, the spatial resolution is reduced by 2^2 via S_j". Consider the case when a tensor with N channels of spatial resolution M by M is passing through layer j. If S_j is a downsampling operation (i.e. j=3, 21, 69 or 285), then x_tilda_{j+1} will have 4N channels of size M/2 by M/2 while x_{j+1} will still have N channels of size M by M. This means that the addition operation in layer j+1 will be an addition between two tensors of unequal dimension. How is this dealt with for the bijective i-RevNet? Thank you!

---

> ### Author Response · Authors · 2018-01-08
> **Thank you for your interest in the paper.**
>
>
> At a given block, the tensor $\tilde x_j$ can potentially have a different size from $x_j$ without losing in generality. Consequently, as the bottleneck layer  $\mathcal{F}_{j+1}$ is applied to $\tilde x_j$, its output must match the shape of $x_j$. This is done, for instance, by applying an intermediary stride via $\mathcal{F}_{j+1}$ (it is also possible to upsample its output), or increasing/reducing the channel size, accordingly to the size of $x_j$.
>
> In our implementation and for consistency, we followed the approach of the RevNet which down-samples two interlaced blocks at a given depth $j$ (e.g. ${x_j,\tilde x_j}). It means that two successive blocks (and thus also $S_j$ & $S_{j+1}$) of an $i$-RevNet work in concert, in order to have downsampled the signals ${x_{j+2},\tilde x_{j+2}}$ by a factor $2^2$ w.r.t. the depth $j$.
>
> We will add clarification of this to the camera ready. As mentioned in the manuscript, we will also release our code so you can check how this is implemented in detail.
>
> We thank you very much again for your comment!

---

### Public Comment · (anonymous) · 2018-01-15
**Two questions about your paper.**

I am interested in your paper. But I have two questions:

1) As mentioned, features at each layer are decomposed into two parts. So how to decompose the input images? (I don't find the corresponding descriptions in your paper.)

2) The reconstructed sequences x_t in Fig.5 contain lots of noise and are with low visual qualities. Can you explain the reasons?

---

> ### Author Response · Authors · 2018-01-16
> **Thank you very much for your interest.**
>
>
> 1) As mentioned in section 3.1 and detailed in the 4th paragraph of section 3.2, $\tilde{S}$ splits the input into two tensors. In our case, we stick to the choice of Revnets and split the number of input channels in half. You will be able to check how this is done in detail in the code we will release alongside the de-anonymized version of the paper.
>
> 2) Thanks for this question. What is displayed in fig. 5, are not noisy images, but rather precise reconstructions of an interpolation in feature space. Images obtained by this interpolation have no particular reason to look like real images, as their representation suffers from the curse of dimensionality. However, as indicated in the paper, it indeed opens the question about the structure of the feature space.

---

### Public Comment · (anonymous) · 2018-01-31
**Regarding the Size of Tensors in the i-RevNet**

It's mentioned when describing the bijective variant of i-RevNet that "the dimensionality of each layer is constantly equal to [3 * 224 * 224], until the final layer, with channel sizes of 24, 96, 384, 1536." It is also mentioned that "the input is split via tilde{S}, which corresponds to an invertible spatial downsampling of 2^2 that increases the number of channels from 3 to 12." I have a few questions regarding this:

(a) Are x_{0} and tilde{x}_0 obtained by performing the channel-wise split on the 12-channel tensor (i.e. are x_{0} and tilde{x}_0 both tensors with 6 channels)?

(b) If the answer to (a) is yes: I am a bit confused about how parameters are being computed. It is mentioned that b-variant of i-RevNet only contains about 29M parameters. I would like to clarify something I believe I'm misunderstanding in the paper.

Consider the number of weights in the three convolutional layers in the bottleneck when there are 1536 channels:

Conv1x1: (1 * 1 * 1536 * 1536/4) = 589824
Conv3x3: (3 * 3 * 1536/4 * 1536/4) = 1327104
Conv1x1: (1 * 1 * 1536/4 * 1536) = 589824

This means each of the weights in each layer of the network with 1536 channels have about 2.5M parameters. I believe the paper mentioned there are about 15 of these layers in the bijective i-RevNet (from layer 285 to 300), which would mean that just this part already has greater than 29M parameters.

(c) Otherwise, if the answer to (a) was no: how are x_{0} and tilde{x}_0 obtained from the 12 channel tensor and what are their respective dimensions? Does x_{0} = tilde{x}_0? Additionally, what are the dimensions of x_{0} and tilde{x}_0 at layer j=1 and j=2 (after the operator tilde{S} but before the first non-identity S_j at j=3)?

(d) Is the dimension of x_{0} and tilde{x}_0 each constantly equal to 3 * 224 * 224 or is the sum of the dimension of x_{0} and tilde{x}_0 constantly equal to 3 * 224 * 224?

---

> ### Author Response · Authors · 2018-02-01
> **Thanks for your interest.**
>
> This is a confusion between "block" (i.e. \mathcal{F}) and layers, each bottleneck block contains three layers. See the camera ready, where we clarify this in the notation.

---

### Public Comment · (anonymous) · 2018-08-02
**Two Questions about the Paper**

After reading your paper and your code on https://github.com/jhjacobsen/pytorch-i-revnet, I have two questions:
1) It is mentioned in Table 1 that i-RevNet(b) includes 29M parameters. However, in the pre-trained model provided on https://drive.google.com/uc?id=1eQtZIpmtVM0-JDv_2nx3xMuGKjA1beLF&export=download, it includes 119M parameters in 'state_dict'. Why do they mismatch?
2) In section 3.2, it is mentioned that the kernel sizes are respectively 1*1, 3*3, 1*1 in each block Fj. However, they are 3*3, 3*3, 3*3 in your code 'iRevNet.py'. Why do they mismatch?

---

> ### Author Response · Authors · 2018-11-05
> **answer**
>
> Sorry for the late response, I overlooked this.
> You can easily change the model in the repo to be exactly the same as in the paper without a problem.
> The weights we provide are just a slightly different version of the network described in the paper for internal reasons, but all results and conclusions remain the same for this model as well.

---

### Decision · Program_Chairs · 2018-01-29
**ICLR 2018 Conference Acceptance Decision**

**Decision:**

Accept (Poster)

**Comment:**

This paper constructs a variant of deep CNNs which is provably invertible, by replacing spatial pooling with multiple shifted spatial downsampling, and capitalizing on residual layers to define a simple, invertible representation. The authors show that the resulting representation is equally effective at large-scale object classification, opening up a number of interesting questions.

Reviewers agreed this is an strong contribution, despite some comments about the significance of the result; ie, why is invertibility a "surprising" property for learnability, in the sense that F(x) = {x,  phi(x)}, where phi is a standard CNN satisfies both properties: invertible and linear measurements of F producing good classification. All in all, this will be a great contribution to the conference.

---

> ### Author Response · Authors · 2018-01-30
> **-**
>
> Dear PCs,
>
> Would it be possible you correct your comment, as in the example you give, $F:x\rightarrow \{x,f(x)\}$ is trivially not invertible for any non constant $f$? Indeed, it is bijective on its image but this is not an interesting property, as you observed. For example, this $F$ can not be used to perform linear interpolation in the feature space as we did.
>
> Invertible means bijective, i.e. injective and surjective and that you need to specify the output space. In the setting you propose, if the input space is $R^d$, then the output space would be $R^{d'}$ with $d'>d$. In this case, if $\exists x_0, x_1, f(x_0)\neq f(x_1)$ then the function you propose is not invertible (what is the preimage of $\{x_0,f(x_1)\}$?), but simply injective. Injectivity is quite trivial to achieve, as you observed.
>
> As you are not surprised, we would be happy and very thankful if you could describe us trivial invertible (or bijective from $R^d$ to $R^d$) CNNs because it would be very helpful for our current research.
>
> Best regards,
> Edouard Oyallon

---

> > ### Comment · Area_Chair · 2018-01-30
> > **clarifications**
> >
> > Hi! First, I hope you did not consider my comment as a critique, but rather as an encouragement to clarify this in your camera-ready version. I really liked the paper!
> >
> > Indeed, this comes down to what we mean by 'invertible'.
> >
> > I took as definition what is suggested by your abstract, "(...) loss of information is not a necessary condition to learn representations that generalize well on complicated problems (...)". We both agree that injectivity alone is not an interesting property!
> >
> > So let me try to precise my previous comment. The invertibility (now in the strict bijective sense) of the final mapping is a consequence of the invertibility of each scale-specific transform from eq(1) (and is independent of F), whereas the generalization ability of the final mapping a priori depends mostly on F (since this is where the learning happens). My question then is: to what extent this phenomena requires several scales (iteration over j) to emerge?
> >
> > Also, now that I look in more detail at eq. (1), unless I am mistaken, there seems to be a size inconsistency: If I understood correctly, in your setup, F is a convolutional net that preserves spatial resolution (no downsampling). In that case, the second equation implies that x_j, F(tilde{x}_j) and therefore tilde{x_j} and tilde{x}_{j+1} have the same spatial resolution. But the first equation then contradicts this since x_{j+1} and tilde{x}_j are related by S and thus cannot have the same spatial resolution. Does that mean that F is also performing some downsampling?
> >
> > As for your final question, no, I don't know of any 'trivial' bijective CNN, but here is a conjecture: Take F(x) = ( W x, Phi(x) ), where Phi is a standard CNN and W is a random iid matrix whose size is adjusted so that F : R^N --> R^N. I conjecture that F is bijective with high probability. Obviously F^{-1} does not have the nice construction as your iRevNet, but it may be represented as a proximal operator: given z = (z_1, z_2) in the domain of F, I would be curious to see if min_x || Wx - z_1 ||^2 + || Phi(x) - z_2|| recovers x.

---

> > > ### Author Response · Authors · 2018-01-30
> > > **Answer**
> > >
> > > Hi AC, thanks, we are happy to get your feedback. Let me follow up as well.
> > >
> > > Informally, you might have the idea in mind of "duplicating" the signal by doing a copy. But we avoid the setting F(x) = \{k(x), phi(x)\} where phi is a large CNN and k is any injective function because this makes the information redundant and the CNN could freely optimize the representations while F being invertible on its image. This setting corresponds roughly to your conjecture with the simplified case: k={identity mapping restricted to the whole image minus one pixel}, and phi, the 1-dimensional class output of your CNN.
> > >
> > > Analyzing only the output of an i-RevNet with a single but deep block(i.e. J=1) will not give much insight on the necessity to discard information, in order to create relevant invariants for classification. An i-RevNet with a lot of distinct blocks necessarily creates and propagates those invariants through the cascade, as shown by the progressive linear separability.
> > >
> > > Exact invertibility is thus surprising when starting from an input image, each successive layer is propagated through a shallow block (repeated J times) and it ends into a vector of the same size. And even more surprising, when it leads to good classification performances on ImageNet. We will stress this even more prominently in the camera ready.
> > >
> > > Best regards,
> > > Edouard Oyallon
> > >
> > > NB: Downsampling: indeed, we answered on 08/01/2018's comment that this is not a mistake in the notation and that "an intermediary stride via $\mathcal{F}_{j+1}$" can be incorporated. As promised, we clarify this in the camera ready.